# Mitigating Classifier Dimensional Collapse via Random Masking in Federated Linear Probing

## Abstract

Integrating a pre-trained model into federated learning (FL) is an emergent direction to facilitate the industrial deployment. Federated linear probing (FLP) is a practical paradigm that gains the communication efficiency from FL and generalization from pre-trained model, while still suffering from dimensional collapse that undermines its effectiveness. Dimensional collapse, originating from data heterogeneity, challenges the embedding space construction in FL, leading to suboptimal convergence and generalization. With a generalized frozen embedding extractor, FLP seems to be robust against dimensional collapse. However, in this paper, we emphasize that the dimensional collapse can also be represented in classifier construction, affecting the performance of the model. We propose FedRM to solve this problem, which randomly masks the dimension of the embedding and the classifier during the training to enforce the classifier to focus fairly on each dimension, guaranteeing diversity during decision generation. The simplicity of the method retains the communication efficiency of the FLP. We conduct empirical experiments to comprehensively evaluate the performance of FedRM. The results show FedRM achieves an overwhelming trade-off between efficiency and utility.

## 1 Introduction

Federated learning ensures that distributed clients collaboratively train a global model while preserving the privacy of their local data McMahan et al. (2017). However, training models from scratch in traditional FL demands high computation and communication costs. To address this issue, recent studies have explored integrating pre-trained models into FL Legate et al. (2023); Tan et al. (2022b). This paradigm not only reduces the high cost of training-from-scratch but also leverages the extensive knowledge encoded in pre-trained models.

Generally, an adapter is concatenated with the pre-trained model to fit in the downstream tasks. From a strategic perspective, FL methods that incorporate pre-trained models can be broadly classified into three categories: zero-shot learning, full-parameter fine-tuning and parameter-efficient fine-tuning, as shown in Fig. 1.

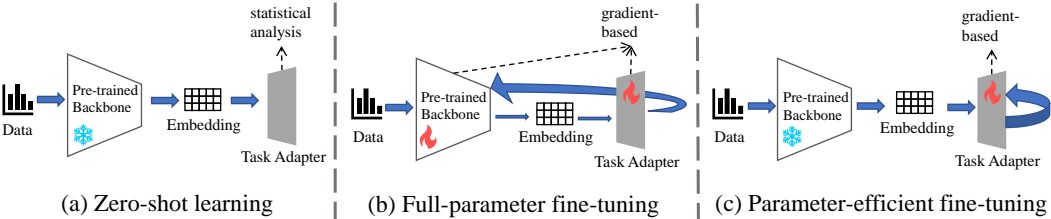

(a) Zero-shot learning   (b) Full-parameter fine-tuning   (c) Parameter-efficient fine-tuning

Figure 1: Illustration of three FL methods with pre-trained model. (a) Zero-shot learning, where task adapter parameters are derived from statistical analysis. (b) Full-parameter fine-tuning, where both the backbone and the adapter are optimized with gradient-based updates. (c) Parameter-efficient fine-tuning, where only a part of parameters are updated through gradient descent.

Namely, zero-shot learning constructs an adapter through statistical analysis without performing gradient-based optimization on local data Legate et al. (2023); He et al. (2025). While zero-shot learning achieves significant training efficiency, it may sacrifice the utility of the model. Conversely, full-parameter fine-tuning utilizes gradient-based optimization to the full parameters of the model for the downstream task. Full-parameter fine-tuning follows the standard FL paradigm, where all model parameters including pre-trained model and adapter are optimized with gradients and communicated across clients. Transmitting full model parameters in each round is costly, motivating the development of parameter-efficient fine-tuning (PEFT) methods in FL Chen et al. (2022); Bian et al. (2025). PEFT optimizes only a subset of critical model parameters with gradients, reducing training cost while maintaining effectiveness. In this paper, we investigate PEFT, as it can achieve a balance between model performance and training efficiency. Specifically, we focus on federated linear probing (FLP), which serves as the most representative one. FLP freezes the shared pre-trained backbone and only trains a linear classifier, enabling lightweight adaptation with reduced computation and communication, while leveraging the generalization of the pre-trained model.

Data heterogeneity, which refers to the divergence between clients' local data distribution, is an inherent challenge of FL leading to suboptimal convergence and generalization Li et al. (2020); Ye et al. (2023). An implicit problem of data heterogeneity is the dimensional collapse in the embedding space, where the representation dimensions are not fully exploited to distinguish different classes. Seemingly, a naive advantage of pre-trained model is to guarantee the quality of embeddings avoiding dimensional collapse. However, in this paper, we emphasize that the dimensional collapse can also occur in the classification layers. The imbalanced local data distribution biases the classifier to concentrate on partial dimensions leading to poor generalization of the local model and causing negative impact on the global model.

To solve this problem, we propose FedRM, which randomly masks the dimensions of the embeddings and the classifier during clients' local training, so that the classifier can attend to different subsets of dimensions rather than only rely on a few dominant directions. FedRM generates multiple masks of different proportions randomly. Both the embedding and the classifier employ these masks to retain different subsets of dimensions and then calculate the regularization separately. By enforcing the classifier to focus on all dimensions evenly, FedRM encourages fairer utilization of the dimensions, thus helping to reduce the risk of dimensional collapse in the classifier. We conduct comprehensive evaluations of FedRM and experimental results show that FedRM outperforms baseline methods. In particular, FedRM performs more effectively on more challenging datasets.

In summary, our main contributions are as follows:

• We focus on FL with pre-trained models, and identify that the dimensional collapse within the classifier is represented as a key challenge in FLP, which negatively impacts the effectiveness in adapting to downstream tasks and ultimately leads to degraded model performance.

• We propose FedRM, a simple and effective method that encourages the classifier to attend to all dimensions of embeddings rather than relying on only a few directions, thereby mitigating dimensional collapse within the classifier and significantly enhancing adaptability in downstream tasks.

• We demonstrate the effectiveness of FedRM through comprehensive experiments on multiple datasets. Moreover, by employing different pre-trained backbones, we show that FedRM can effectively address the problem of classifier dimensional collapse under data heterogeneity.

## 2 RELATED WORK

### 2.1 FEDERATED LEARNING

FL McMahan et al. (2017) has emerged as a promising machine learning paradigm that enables multiple distributed clients to collaboratively train a shared model without exposing or transmitting their raw local data. By keeping data decentralized and only exchanging model updates, FL not only ensures effective preservation of data privacy but also facilitates large-scale collaboration across heterogeneous participants. FedAvg McMahan et al. (2017), which is a foundational algorithm, aggregates the locally trained model parameters from all clients to update a shared global model. However, under data heterogeneity, FedAvg performance deteriorates significantly when faced with data heterogeneity Wang et al. (2020); Dai et al. (2023); Qu et al. (2022).

To tackle this challenge, numerous methods have been proposed to mitigate the adverse effects of heterogeneity. FedProx Li et al. (2020) makes local updates close to the global model by adding a proximal term to the local objective. MOON Li et al. (2021) introduces model-level contrastive learning in FL to align local and global representations. Another research direction for mitigating data heterogeneity is personalized federated learning Tan et al. (2022a); Arivazhagan et al. (2019), which aims to learn a personalized model for each participant that is adapted to its unique local data distribution, while still benefiting from the collective knowledge shared across the federated network. In this paper, we focus on the problem of classifier dimensional collapse caused by data heterogeneity.

## 2.2 DIMENSIONAL COLLAPSE

Dimensional collapse is the phenomenon where representations occupy only a low-dimensional subspace within a high-dimensional feature space, thereby limiting their expressive ability Jing et al. (2021). In FL, Shi et al. (2022) observes that data heterogeneity induces dimensional collapse in both local and global models at the representation level. In this paper, however, we investigate dimensional collapse occurring in the classifier in FLP, where data heterogeneity causes the classifier to rely on only a few dimensions rather than attending uniformly to all dimensions.

## 2.3 FL WITH PRE-TRAINED MODELS

In FL with pre-trained models, adapting the model to downstream tasks can follow different approaches based on the balance between accuracy and communication overhead. Existing approaches can be categorized into zero-shot learning, full-parameter fine-tuning, and parameter-efficient fine-tuning.

Zero-shot learning directly applies the pre-trained model to downstream tasks without any training. FedNCM Legate et al. (2023) aggregates local class means into global class means then uses them to perform classification directly or initialize classifier weights without any training. Fed3R Fanì et al. (2024) solves the ridge regression problem in FL through sharing second-order feature statistics to obtain the closed-form solution classifier. In AFL He et al. (2025), each client shares a common frozen pre-trained backbone to extract embeddings and undergoes only one epoch of training to obtain an analytical solution. The server uses the absolute aggregation law to aggregate the results in a single round of communication. The gradient-free design of AFL reduces overhead, enabling scalability to resource-limited or large-scale FL scenarios. However, zero-shot inference lacks task-specific parameter adaptation, which may lead to suboptimal performance in some specific scenarios.

Full-parameter fine-tuning addresses this issue by updating all model parameters, enabling thorough adaptation to the downstream task. FedKSeed Qin et al. (2023) utilizes zeroth-order optimization with a finite set of $K$ random seeds and scalar gradients in federated full-parameter tuning, enabling it to maintain competitive accuracy while significantly reducing communication costs. Ferret Shu et al. (2024) achieves the federated full-parameter tuning through low-dimensional projection and reconstruction techniques, introducing the first-order method with shared randomness. Nevertheless, transmitting the full parameters in each communication round incurs significant overhead. To tackle the issue, many methods have proposed parameter-efficient fine-tuning in FL.

Parameter-efficient fine-tuning methods, including adapter tuning, LoRA, and prompt tuning, only update a small subset of parameters while keeping the majority of the model fixed. FedPCL Tan et al. (2022b) leverages prototype contrastive learning to share class prototypes between clients, thereby enabling knowledge fusion from multiple pre-trained models. PROMPTFL Guo et al. (2023) enables federated participants to cooperatively train prompts instead of full models, leveraging a frozen backbone like CLIP to achieve efficient aggregation, reduced communication overhead, and improved performance with limited local data.

## 3 BACKGROUND

### 3.1 FEDERATED LEARNING

FL allows distributed clients to collaboratively train a deep learning model without direct data communication. Considering a FL system targeting classification tasks with $K$ clients, the $k$-th client maintains a local dataset denoted as $\mathcal{D}_k = \{(X_{k,i}, y_{k,i})\}_{i=1}^{N_k}$, where $X_{k,i}$ and $y_{k,i}$ represent the $i$-th feature-label pair, and $N_k$ is the number of data samples. The global objective of FL is a weighted sum of the local objectives of clients which can be defined as:

$$\min_{\delta} \sum_{i=1}^{K} \alpha_k \mathcal{L}_k(\delta; \mathcal{D}_k), \tag{1}$$

where $\mathcal{L}$ and $\delta$ represent the local objective and the model respectively. $\alpha_k$ denotes the weight of client $k$. Further we define the local objective of the $k$-the client as follows:

$$\mathcal{L}_k(\delta; \mathcal{D}_k) = \mathop{\mathbb{E}}_{(X_{k,i}, y_{k,i}) \sim \mathcal{D}_k} [\mathcal{L}_{CE}(W(\delta; X_{k,i}), y_{k,i})], \tag{2}$$

where $W$ is the mapping function which maps the input $X$ to its corresponding output through model $\delta$. $\mathcal{L}_{CE}$ is the Cross Entropy loss function which calculates the divergence between model output and the true label.

The FL system iteratively executes the following steps to solve the global optimization problem. Specifically, the server first sends the global model parameters to a subset of clients. The sampled clients load the model parameters to their local models and perform local training on their dataset. After local training, the clients send the local model back to the server. Finally, the server performs model aggregation on the received local models to generate a global model for the next iteration of training.

### 3.2 FL WITH PRE-TRAINED MODELS

With the development of pre-training technology, a new trend of research has integrated pre-trained models into FL. Specifically, in contrast to traditional FL trained from scratch, the global model can be initialized with a pre-trained model concatenated with an adapter fitting downstream tasks. The model $\delta$ and the mapping function $W$ can be further specified as:

$$\delta = \theta \oplus \varphi, \tag{3}$$

$$W(\delta, X) = \mathcal{G}(\varphi; \mathcal{F}(\theta; X)), \tag{4}$$

where $\theta$ is a pre-trained backbone and $\varphi$ is a task adapter to the downstream tasks. $\mathcal{G}$ and $\mathcal{F}$ are mapping functions for $\theta$ and $\varphi$ respectively. Strategically, we divide the FL methods integrating with pre-trained model into three categories:

**Zero-shot Learning**

Zero-shot learning emphasizes to solve the problem via statistical analysis without gradient-based optimization method. Formally the global optimization problem is defined as follows:

$$\min_{\varphi} \sum_{i=1}^{K} \alpha_k \mathcal{L}_k(\theta \oplus \varphi; \mathcal{D}_k), \tag{5}$$

where these methods search for the best task adapter that fits the task. In zero-shot learning, the optimization method is always based on statistical analysis. These methods can find the optimization solution through single round communication without gradient-based training on local data.

**Full-parameter Fine-tuning**

Full-parameter fine-tuning utilizes the traditional FL paradigm performing gradient-based optimization method on the entire model parameters to solve the problem. The global optimization problem is defined as:

$$\min_{\theta;\varphi} \sum_{i=1}^{K} \alpha_k \mathcal{L}_k(\theta \oplus \varphi; \mathcal{D}_k). \tag{6}$$

The methods update and communicate the entire model parameters through FL system to solve the optimization problem.

**Parameter-efficient Fine-tuning**

Parameter-efficient fine-tuning leverages gradient-based optimization methods and optimizes only a part of model parameters to solve the problem. Its optimization problem can be defined as:

$$\min_{P(\theta;\varphi)} \sum_{i=1}^{K} \alpha_k \mathcal{L}_k(\theta \oplus \varphi; \mathcal{D}_k), \tag{7}$$

where, $P(\theta;\varphi)$ selects the critical parameters which should be optimized to solve the problem.

### 3.3 Federated Linear Probing

In this paper, we focus on Federated Linear Probing (FLP) which is a typical PEFT method exploited by academic researches and industrial applications. For each client $k$, model $\delta_k$ consists of two components: a shared frozen pre-trained backbone $\mathcal{F}$ parameterized by $\theta$ and a linear layer serving as the linear classifier $\mathcal{G}$ parameterized by $\varphi_k$. The goal of FLP is to train the linear classifier of the downstream tasks in FL paradigm. With the backbone frozen, the embedding of the data can be extracted at the initialization of the learning and reused during the iterations which improve the efficiency of the training.

Each client $k$ extracts embeddings $\mathcal{R}_{k,i}$ of the data point $X_{k,i}$ using the frozen pre-trained backbone network $\mathcal{F}$ as follows:

$$\mathcal{R}_{k,i} = \mathcal{F}(\theta; X_{k,i}), \tag{8}$$

where $\mathcal{R}_{k,i} \in \mathbb{R}^{1 \times d}$, and the embedding has a total of dimensions $d$, denoted as the set $\mathcal{I} = \{1, 2, ..., d\}$. The local objective of each client $k$ is as follows:

$$\mathcal{L}_k(\delta_k; \mathcal{D}_k) = \mathbb{E}_{(X_{k,i}, y_{k,i}) \sim \mathcal{D}_k} [\mathcal{L}_{CE}(\mathcal{G}(\varphi_k; \mathcal{R}_{k,i}), y_{k,i})]. \tag{9}$$

In each round of the training, the server only communicates and aggregates the parameters of the classifier. The clients also update the parameters of the classifier based on their local objectives.

## 4 Methodology

In this section, we discuss the issues of local and global dimensional collapse induced by data heterogeneity in Sec. 4.1, particularly in classifiers. To address this issue, we propose FedRM in Sec. 4.2, which leverages multi-ratio random masking.

### 4.1 Dimensional Collapse in Classifier

In FL, the data distribution across clients is typically not independent and identically distributed (Non-IID). Clients with limited class diversity often develop restricted embedding spaces in their local models, where embeddings are concentrated along only a few dominant directions. This phenomenon reduces the effective dimensionality of the representation space and ultimately leads to dimensional collapse in local embeddings.

With the introduction of pre-trained models into FL, the rich knowledge embedded in these models can be generalized to the downstream tasks to alleviate dimensional collapse at the embedding level. Recent studies have demonstrated the effectiveness of pre-trained models against Non-IID in FL Legate et al. (2023); Weng et al. (2024); He et al. (2025).

However, in the case of FLP, the training of the classifier still faces the challenge of Non-IID. We identify that the training of a linear classifier can still be affected by dimensional collapse. Due to data heterogeneity, the local data distribution of clients can be biased to certain classes. Consequently, the local classifier only needs to learn to distinguish in these few low-dimensional directions, resulting in significant changes in these directions, and the classifier gradually collapses to the low-dimensional subspace that distinguishes the minority class.

Furthermore, since the global classifier is aggregated from the local classifiers, it inevitably inherits the dimensional collapse occurring at the local level. As training progresses, the global classifier aggregates multiple times, causing the dimensional collapse within the global classifier and limiting its generalization ability on various downstream tasks.

## 4.2 FEDRM

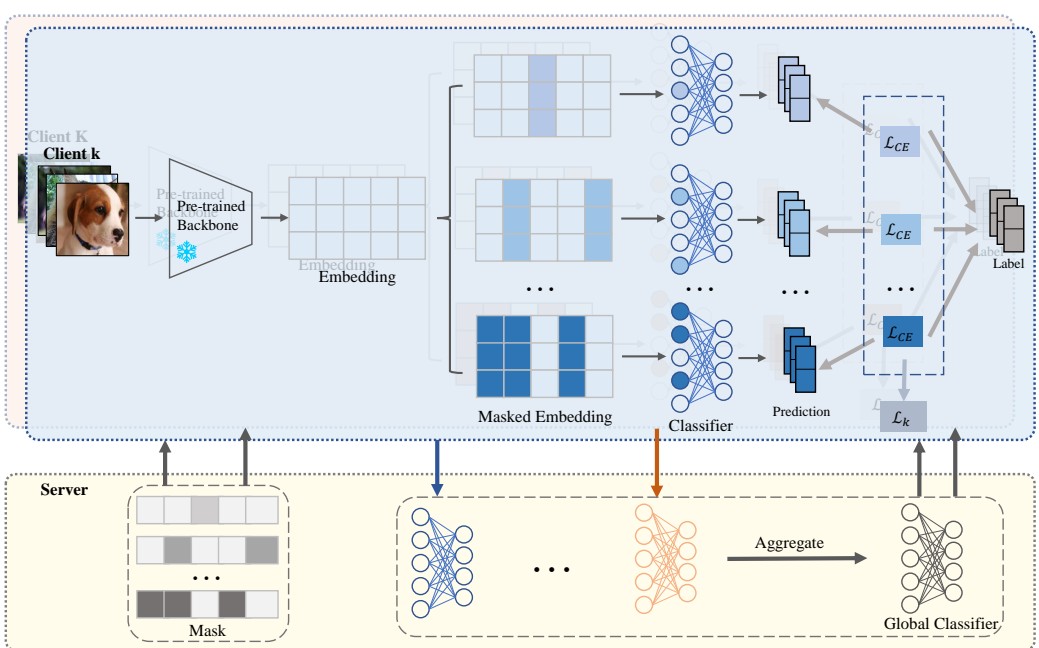

Figure 2: Framework of FedRM. The server generates random masks and distributes it to each client. The client uses the masks to obtain the corresponding masked embeddings and classifiers, so that the classifier does not rely solely on a few dimensions.

To overcome dimensional collapse in the classifier, we propose FedRM which can be treated as a regularization introduced to the local training of FLP, with the framework illustrated in Fig. 2. Specifically, FedRM applies $M$ different masks with varying masking ratios to the embedding, generating a set of representations with different effective dimensions. For each masked representation, the corresponding regularization is computed, and the total regularization is obtained by summing all $M$ masked versions.

Let $\mathcal{M}^{(m)} \subset \mathcal{I}$ be the subset of dimensions that are masked by the $m$-th mask, which satisfies:

$$|\mathcal{M}^{(m)}| = \left\lfloor d \cdot r^{(m)} \right\rfloor \ and \ \mathbb{P}(\mathcal{M}^{(m)} = S) = \frac{1}{\binom{d}{|S|}}, \forall S \subset \mathcal{I, \quad (10)$$

where $r^{(m)}$ is the percentage of the $m$-th mask and $|\mathcal{M}^{(m)}|$ is the number of dimensions contained in this subset. The $\mathcal{M}^{(m)}$ is sampled uniformly at random from all possible subsets, without re-

placement. Define the mask vector $mask^{(m)} \in \{0,1\}^d$ corresponding to the $m$-th mask such that:

$$mask_i^{(m)} = \begin{cases} 0, & \text{if } i \in \mathcal{M}^{(m)} \\ 1, & \text{otherwise} \end{cases} \tag{11}$$

where 0 indicates that the dimension is masked, while 1 indicates that it is retained. For the $m$-th mask, a corresponding masked representation and classifier are used to compute the associated regularization. The regularization function of FedRM for client $k$ is as follows:

$$\mathcal{L}_{RM}(\varphi_k, \mathcal{D}_k) = \mathbb{E}_{(X_{k,i}, y_{k,i}) \sim \mathcal{D}_k} \sum_{m=1}^{M} \mathcal{L}_{CE}(\mathcal{G}(mask^{(m)} \odot \varphi_k; mask^{(m)} \odot \mathcal{R}_{k,i}), y_{k,i}), \tag{12}$$

where $\odot$ represents the Hadamard product. Adding different masks to the embedding forces the classifier to use multiple subspaces, avoiding using only a few directions, thereby solving the dimensional collapse of the classifier. Further we define the loss function of the client as follows:

$$\mathcal{L}_k(\varphi_k, \mathcal{D}_k) = \mathcal{L}_{CE}(\varphi_k, \mathcal{D}_k) + \alpha_{RM} \mathcal{L}_{RM}(\varphi_k, \mathcal{D}_k), \tag{13}$$

where $\mathcal{L}_{RM}$ is weighted by a factor $\alpha_{RM}$ and added to the local objective of the clients.

Upon completion of local training, each client uploads its classifier parameters $\varphi_k$ to the server, where they are aggregated by averaging as follows:

$$\varphi = \frac{N_k}{N} \sum_{i=1}^{K} \varphi_k, \tag{14}$$

where $\varphi$ represents the global classifier and $N$ is the total number of data.

The core reason supporting FedRM to conquer dimensional collapse is the employment of multiple random masking of classifier dimensions. Random masking enforces the classifier to concentrate on specific dimensions during the decision making avoiding dependence on the easy dimensions. Multiple masks guarantee the diversity of the dimensions exploited in classification enhancing the generalization of the classifier.

## 5 EMPIRICAL EVALUATION

### 5.1 EXPERIMENT SETTING

**Datasets.** Our experiments involve the following datasets: Caltech256 Griffin et al. (2007), PathMNIST Yang et al. (2023), EuroSAT Helber et al. (2019), CIFAR-100 Krizhevsky et al. (2009), TinyImageNet Le & Yang (2015), Flower102 Nilsback & Zisserman (2008), StanfordCars Krause et al. (2013) and iNaturalist (2021 train mini) Van Horn et al. (2018). To construct scenarios with heterogeneous clients, we use the Dirichlet distribution $Dir(\beta)$ to partition the dataset and set $\beta \in \{0.05, 0.1\}$.

**Pre-trained model.** In our experiments, we mainly adopt two widely used pre-trained backbones: namely CLIP Radford et al. (2021) and ViT-Base Dosovitskiy et al. (2020), which are employed as powerful feature extractors to generate high-quality embeddings from raw input data, where CLIP is pre-trained on publicly available image-caption data and ViT-Base is pre-trained on the ImageNet-21k dataset. To focus our study on the behavior of the classifier, we keep the parameters of the pre-trained backbone frozen throughout the entire training process. This allows us to leverage the strong representation capability of pre-trained models while ensuring that any performance gains can be attributed to the improvements brought by FedRM itself. Due to space limitation, we only show the experiment results of CLIP in our main paper. The results of ViT-Base are referred to the appendix of the paper.

**FL setting.** Our experimental setup involves a total of 100 clients, where in each communication round only 10% of the clients are randomly selected to participate. For most datasets, we train the global model for 100 communication rounds, while for the more challenging iNaturalist dataset, we extend the training to 200 rounds in order to ensure sufficient convergence. We use a batch size of 64 and adopt the SGD optimizer with weight decay 1e-5 and momentum 0.9. The local update is

performed for 5 epochs with a learning rate of 0.005. All experiments are implemented in PyTorch 2.7.1 running on NVIDIA A100.

**Baselines.** We compare our method FedRM with multiple FL algorithms that leverage pre-trained models. The baselines involving zero-shot learning include FedNCM Legate et al. (2023) and Fed3R Fanì et al. (2024). In addition, when the pre-trained model is CLIP Radford et al. (2021), we additionally evaluate its direct zero-shot inference capability as a baseline. For parameter-efficient fine-tuning, we include FedAvg McMahan et al. (2017), Fed3R+FTLP Fanì et al. (2024) and FedMRL. Specifically, Fed3R+FTLP initializes the classifier using Fed3R and subsequently fine-tunes it to adapt to downstream tasks. FedMRL integrates matryoshka representation learning Kusupati et al. (2022) into the FedAvg framework. Instead of learning a single flat representation, the model is trained to generate a nested set of representations at different levels of granularity.

## 5.2 MAIN RESULTS

In this experiment, we validate the effectiveness of FedRM under different heterogeneous settings, where the degree of heterogeneity across clients is controlled by the Dirichlet coefficient $\beta \in \{0.05, 0.1\}$. To ensure a fair comparison and stabilize the optimization process, we set the regularization coefficient to $\alpha_{RM} = 1$, while employing CLIP as the pre-trained backbone to provide powerful and generalizable feature representations.

Table 1 reports the results of this experiment. FedRM consistently achieves the highest accuracy compared with both zero-shot inference and PEFT methods. This demonstrates that, in the presence of data heterogeneity, FedRM effectively alleviates dimensional collapse in the classifier. By applying diverse random masks to the embedding, FedRM encourages the classifier to exploit the full range of feature dimensions rather than relying only on a limited set of discriminative directions. Such a strategy enables more balanced utilization of the representation space, thereby improving both robustness and generalization across clients.

Table 1: Test accuracy(%) on three datasets CIFAR-100, Caltech256 and iNaturalist under the Non-IID setting, with CLIP as the pre-trained backbone. The best performance is overstriking in the table

| Category | Method | CIFAR-100 | | Caltech256 | | iNaturalist | |
|---|---|---|---|---|---|---|---|
| | | $\beta$=0.05 | $\beta$=0.1 | $\beta$=0.05 | $\beta$=0.1 | $\beta$=0.05 | $\beta$=0.1 |
| | FedNCM | 85.34 | 85.19 | 63.46 | 63.51 | 20.21 | 20.21 |
| Zero-shot | Fed3R | 79.33 | 17.13 | 63.98 | 36.73 | 17.32 | 17.91 |
| | CLIP-zero-shot | 79.55 | 79.48 | 60.23 | 60.23 | 1.93 | 1.93 |
| | FedAvg | 82.40 | 82.30 | 65.53 | 66.73 | 8.99 | 4.13 |
| PEFT | Fed3R+FTLP | 86.28 | 83.03 | 67.39 | 67.44 | 13.67 | 13.78 |
| | FedMRL | 84.89 | 84.55 | 64.45 | 66.03 | 13.22 | 13.25 |
| | FedRM | **87.32** | **87.64** | **68.59** | **70.65** | **21.45** | **21.64** |

We conduct comparative experiments between the baseline method FedAvg and our proposed FedRM across multiple datasets. In these experiments, we set the Dirichlet coefficient to $\beta = 0.05$ to simulate a high degree of data heterogeneity, and employ CLIP as the pre-trained backbone.

As shown in Table 2, the experimental results reveal that as the complexity of the datasets increases, the problem of dimensional collapse in the classifier becomes more pronounced. Notably, FedRM demonstrates increasingly significant performance improvements on more challenging datasets, indicating its superior ability to effectively mitigate dimensional collapse. These findings highlight the robustness and excellence of FedRM in handling heterogeneous and complex FL scenarios.

## 5.3 ABLATION STUDY

To further investigate the effectiveness of multi-ratio random masks in the FedRM framework, we conduct a series of ablation experiments on the Caltech256 dataset under the heterogeneous setting with Dirichlet coefficient $\beta = 0.1$. In FedRM, multiple masks with ratios ranging from 10% to 90% are applied to the embedding space. We construct two sets of FedRM variants by removing one

Table 2: Comparison of test accuracy(%) between FedAvg and FedRM across multiple datasets, with CLIP as the pre-trained backbone and $\beta = 0.05$. The best performance is overstriking in the table.

| Method | Caltech256 | PathMNIST | EuroSAT | CIFAR-100 |
|--------|-----------|-----------|---------|-----------|
| FedAvg | 82.40 | 73.93 | 67.85 | 65.53 |
| FedRM | **87.32** | **77.92** | **69.83** | **68.59** |

| Method | TinyImageNet | Flower102 | StanfordCars | iNaturalist |
|--------|--------------|-----------|--------------|-------------|
| FedAvg | 61.13 | 37.94 | 38.23 | 8.99 |
| FedRM | **65.37** | **63.49** | **58.02** | **21.45** |

mask at a time in descending or ascending order, thereby examining how the absence of mask ratio influences the overall performance and robustness of the framework.

As illustrated in Fig. 3a, the masks are decremented in descending order of their ratios from 90% to 10%. Fig. 3b depicts the scenario where masks are decremented in ascending order of ratios from 10% to 90%. The text within the brackets in the legend indicates the ratio ranges corresponding to the masks of each variant. The results of the ablation study demonstrate that as the number of random masks is sequentially reduced, the performance of FedRM gradually degrades. This degradation underscores the importance of maintaining a diverse set of masks, as such diversity allows FedRM to more comprehensively attend to all dimensions of the embedding space, thereby preventing the classifier from relying on only a few discriminative directions. By encouraging balanced utilization of feature dimensions, the presence of multiple masks ensures more robust learning and better generalization under data heterogeneity, thereby addressing the classifier dimensional collapse problem in FLP. These findings indicate that the multi-ratio random masking strategy plays an important role.

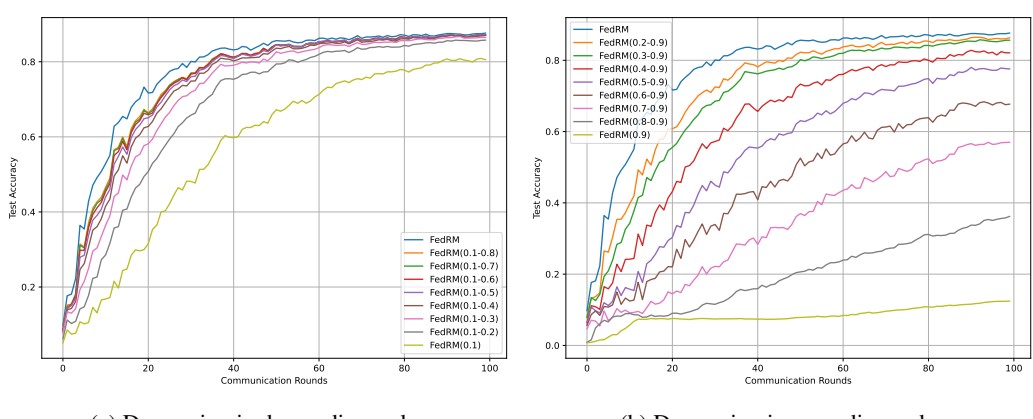

(a) Decreasing in descending order    (b) Decreasing in ascending order

Figure 3: Ablation experiment on the Caltech256 dataset, with a Dirichlet coefficient of 0.1 and a pre-trained backbone of CLIP

## 6    CONCLUSION

In this work, we propose FedRM, a simple but effective framework designed to mitigate classifier dimensional collapse in FLP under data heterogeneity. Specifically, FedRM employs multiple random masks with varying ratios applied to the embedding, compelling the classifier to attend to all feature dimensions. This method sums up the corresponding mask regularizations to jointly guide the optimization process, enhancing the adaptability of the pre-trained model to downstream tasks. Moreover, FedRM achieves a favorable balance between communication efficiency and model utility. Extensive experimental results across multiple datasets confirm the superiority of FedRM over existing baselines, with its performance gains becoming increasingly significant as the complexity of the dataset grows.

## REPRODUCIBILITY STATEMENT

All source code will be publicly available upon acceptance of the paper. In this paper, we describe all baselines and experimental settings in Sec. 5.1, including hyperparameter settings, optimizer configuration, and dataset partitioning strategies. We also specify the computational environment used in the experiments. A detailed description of our proposed method is provided in Sec. 4.2.

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

# A SUPPLEMENTARY RELATED WORK

We classify the methods that integrate FL with pre-trained models into three categories: zero-shot learning, full-parameter fine-tuning, and parameter-efficient fine-tuning. A summary of the relevant literature is provided in Table 3.

Table 3: FL with pre-trained models

| Category | Method | Method Highlight |
|---|---|---|
| **Zero-shot learning** | FedNCM Legate et al. (2023) | Class mean aggregation for training-free classification |
| | Fed3R Fanì et al. (2024) | Second-order statistics sharing for solving the FL ridge regression |
| | AFL He et al. (2025) | Gradient-free and one-epoch local training FL method with absolute aggregation |
| | FedCOF Goswami et al. (2024) | Training-free FL method that estimates class covariances from shared class means |
| **Full-parameter fine-tuning** | FedKSeed Qin et al. (2023) | Zeroth-order optimization with random seeds and scalar gradients |
| | Ferret Shu et al. (2024) | Low-dimensional projection and reconstruction introducing the first-order method |
| **Parameter-efficient fine-tuning** | FedPCL Tan et al. (2022b) | Federated prototype-wise contrastive learning |
| | PROMPTFL Guo et al. (2023) | Collaborative prompt learning for FL |

# B SUPPLEMENTARY EXPERIMENTS

## B.1 EXPERIMENTS WITH VIT-BASE AS THE PRE-TRAINED BACKBONE

In this experiment, we validate the effectiveness of FedRM under different heterogeneous settings, where the data distribution among clients is controlled by the Dirichlet coefficient $\beta \in \{0.05, 0.1\}$.

Table 4 reports the results using ViT-Base as the pre-trained backbone. The experimental results indicate that while FedRM delivers competitive performance, it is slightly outperformed by Fed3R+FTLP. Specifically, Fed3R+FTLP first employs the zero-shot method Fed3R to initialize the classifier and subsequently applies PEFT, thereby equipping the classifier with a superior initialization. While FedRM does not employ this initialization strategy, its performance remains comparable to that of this approach, combining zero-shot learning with PEFT.

Table 4: Test accuracy(%) on three datasets CIFAR-100, Caltech256 and iNaturalist under the Non-IID setting, with ViT-Base as the pre-trained backbone. The best performance is overstriking in the table

| Category | Method | CIFAR-100 | | Caltech256 | | iNaturalist | |
|---|---|---|---|---|---|---|---|
| | | $\beta$=0.05 | $\beta$=0.1 | $\beta$=0.05 | $\beta$=0.1 | $\beta$=0.05 | $\beta$=0.1 |
| Zero-shot | FedNCM | 87.29 | 87.32 | 71.98 | 72.20 | / | / |
| | Fed3R | 79.00 | 78.56 | 66.10 | 66.84 | 8.38 | 9.06 |
| PEFT | FedAvg | 88.89 | 89.29 | 77.58 | 78.15 | 37.79 | 38.42 |
| | Fed3R+FTLP | **89.67** | **89.81** | **78.01** | **78.56** | **39.25** | **39.47** |
| | FedMRL | 87.78 | 87.94 | 74.17 | 74.47 | 36.83 | 37.11 |
| | FedRM | 89.24 | 89.47 | 77.15 | 77.92 | 39.17 | 39.32 |

## B.2 COMPARISON BETWEEN FEDAVG AND FEDRM ON MULTIPLE DATASETS

We conduct comparative experiments between the baseline method FedAvg and our proposed FedRM across multiple datasets. In these experiments, we also set the Dirichlet coefficient to $\beta = 0.1$ to simulate a high degree of data heterogeneity, and employ CLIP as the pre-trained backbone.

As shown in Table 5, experimental results show that the classifier dimensional collapse problem becomes more prominent as dataset complexity increases. FedRM shows increasingly significant performance improvements on more challenging datasets, with only a performance degradation on the PathMNIST dataset.

Table 5: Comparison of test accuracy(%) between FedAvg and FedRM across multiple datasets, with CLIP as the pre-trained backbone and $\beta = 0.1$. The best performance is overstriking in the table.

| Method | Caltech256 | EuroSAT | CIFAR-100 | TinyImageNet |
|--------|-----------|---------|-----------|--------------|
| FedAvg | 82.30 | 75.46 | 66.73 | 62.32 |
| FedRM | **87.64** | **79.93** | **70.65** | **66.04** |

| Method | PathMNIST | Flower102 | StanfordCars | iNaturalist |
|--------|-----------|-----------|--------------|-------------|
| FedAvg | **65.49** | 40.43 | 38.91 | 9.15 |
| FedRM | 51.66 | **58.11** | **58.48** | **21.64** |

## C THE USE OF LARGE LANGUAGE MODELS

In this work, Large Language Models were only used as assist tools for minor language polishing and grammar improvement. They did not contribute to research ideation, methodology design, analysis, or writing of substantive scientific content.

