# OpenReview forum: "Mitigating Classifier Dimensional Collapse via Random Masking in Federated Linear Probing"
_ICLR.cc/2026/Conference — ICLR 2026 Conference Withdrawn Submission_

### Official Review · Reviewer_7jxB · 2025-10-29

**Soundness:** 2
**Presentation:** 3
**Contribution:** 2
**Rating:** 2
**Confidence:** 4

**Summary:**

This paper studies dimensional collapse in federated linear probing (FLP) scenarios and proposes FedRM, a method that applies multi-ratio random masking to both the embedding and the classifier during training. By enforcing attention to diverse subsets of feature dimensions, FedRM aims to avoid over-reliance on a few discriminative directions, promoting more robust classifier generalization under data heterogeneity. The method is lightweight and communication-efficient. Extensive experiments are conducted on various image classification benchmarks under heterogeneous federated settings.

**Strengths:**

1）The paper identifies an overlooked problem in federated learning, dimensional collapse at the classifier level, even when embeddings from frozen pre-trained backbones are robust, highlighting an important gap in the current literature.

2）FedRM is straightforward to implement and computationally lightweight, and can be seamlessly integrated into existing methods. Furthermore, the overhead introduced by this method is minimal, making it highly efficient.

3）The paper is clearly written and easy to follow. Additionally, Figure 2 provides an intuitive overview of FedRM, helping readers grasp the workflow and the masking scheme.

**Weaknesses:**

1）The paper's core premise focuses on FLP, which is an increasingly outdated approach. The current state-of-the-art in federated fine-tuning predominantly leverages Parameter-Efficient Fine-Tuning (PEFT) methods, such as LoRA or Adapters, which adapt the model's internal representations rather than just the final layer. It is widely acknowledged that adapting only the final output layer often yields suboptimal performance, thus limiting the practical utility of the proposed method.

2）The paper's central claim—"dimensional collapse in the classifier" (Section 4.1)—is poorly substantiated. The analysis is superficial, relying on brief textual descriptions rather than rigorous theoretical or experimental evidence. Despite motivating the entire work from this perspective, the paper lacks deep theoretical insights. No formal definitions or quantification (e.g., metrics reflecting the effective dimensionality or a direct analysis of the classifier weights' singular value distribution) are provided to empirically demonstrate the existence or severity of this challenge.

3） The proposed FedRM method offers limited novelty, as it closely resembles a standard regularization technique (e.g., Dropout) applied to the embeddings and classifier weights. Additionally, the experimental evaluation is confined to traditional classification tasks. This overlooks the current, dominant trend of applying pre-trained models (such as LLMs and MLLMs) to more complex and relevant generative and understanding tasks.

4）The choice of baselines is weak and lacks diversity. The paper fails to benchmark against modern, relevant state-of-the-art methods in federated PEFT (e.g., FLoRA[1]). Furthermore, there is no direct benchmarking or discussion of other highly relevant recent approaches that also employ masking, random noise, or collapse mitigation strategies in federated learning.



[1] FLoRA: Federated Fine-Tuning Large Language Models with Heterogeneous Low-Rank Adaptations

**Questions:**

See weakness.

---

### Official Review · Reviewer_7ofs · 2025-10-30

**Soundness:** 1
**Presentation:** 2
**Contribution:** 1
**Rating:** 2
**Confidence:** 3

**Summary:**

This paper addresses the dimensional collapse presented in the classifier due to data heterogeneity. The proposed method, FedRM, introduces random masks during training to enforce the classifier to focus fairly on each dimension. Extensive experiments under various models and datasets demonstrate that the proposed method achieves an overwhelming trade-off between efficiency and utility. However, the score of this paper tends toward rejection because: (1) the core claim regarding classifier dimensional collapse is not well justified both theoretically and empirically, and (2) the experimental claims do not fully support the contribution of the proposed method.

**Strengths:**

This paper addresses the data heterogeneity problem in federated linear probing through simple yet effective modifications. Extensive experiments across various models and datasets demonstrate that the proposed method achieves an excellent trade-off between efficiency and utility.

**Weaknesses:**

(1) The core claim regarding classifier dimensional collapse is not well justified both theoretically and empirically. To address this concern, please clarify and provide detailed explanations for the following:

(1-a) A critical gap in the analysis is how FedAvg's aggregation mechanism interacts with local dimensional collapse. When multiple clients with different biased classifiers aggregate their parameters, the averaging process naturally generalizes dimension-specific biases across clients. The paper claims that “the global classifier inevitably inherits the dimensional collapse” (lines 280–282) without addressing why FedAvg's generalization does not mitigate this issue. Please clarify: What specifically differentiates the collapse mitigation achieved by FedRM from the natural mitigation effect of FedAvg aggregation? Can you provide a comparative analysis regarding this concern? For example, showing dimension utilization patterns for (i) local classifiers before aggregation, (ii) the global classifier after FedAvg, and (iii) the global classifier with FedRM. This is particularly important since the Personalized FL literature suggests that simple aggregation like FedAvg can diminish individual client characteristics under data heterogeneity, which would naturally counteract dimension-specific biases.

(1-b) The paper lacks a theoretical analysis explaining why uniform dimension utilization is beneficial. Why is forcing the classifier to attend to all dimensions uniformly better than allowing it to focus on task-relevant dimensions? Could enforcing uniform utilization harm performance when certain dimensions are genuinely more informative for the task? Please provide a clear and intuitive explanation, preferably with empirical support.

(1-c) The paper claims that “we investigate dimensional collapse occurring in the classifier in FLP, where data heterogeneity causes the classifier to rely on only a few dimensions rather than attending uniformly to all dimensions.” (lines 125–127), yet provides no empirical evidence, such as a visualization of the parameters of the classifier.

(2) The experimental claims do not fully support the contribution of the proposed method. To address this concern, please clarify and provide detailed explanations for the following:

(2-a) All results appear to be from single runs without confidence intervals. For rigorous evaluation, please provide mean ± standard deviation over multiple random seeds.

(2-b) With M masks, the proposed method performs M forward passes per training step. How much does the computational cost increase as the number of masks increases, and how many masks are used as the default?

(2-c) Figure 3 only removes masks incrementally, but critical ablations need to be provided to verify the robustness of the proposed method, such as the effect of the regularization weight $α_{RM}$.

(2- d) Performance comparison with zero-shot methods does not necessarily support the superiority of the proposed method, since those methods are training-free and have fundamentally different computational profiles. If the authors aim to present the superiority of the FLP-like method, a full fine-tuning–based method needs to be added to the comparison.

Also, there are several minor issues to improve the paper:

(3) For reproducibility, it would be better to provide complete algorithm pseudocode showing the full training loop, including mask generation, application, and aggregation.

(4) Figure 2 has multiple issues: the server’s mask section shows two arrows with unclear meaning; the client’s embedding is depicted as a matrix, but Equation (8) (line 248) defines $𝑅_{𝑘,𝑖} ∈ 𝑅^{1×𝑑}$ as a vector. If the matrix represents batched embeddings (batch_size × d), this should be explicitly stated, and the notation should be adjusted throughout.

**Questions:**

Please refer to the Weakness section for detailed comments. In particular, I would appreciate clarification on the questions raised for each weakness. I will reconsider my evaluation after reviewing the authors’ rebuttal to these points.

---

### Official Review · Reviewer_kqJq · 2025-10-30

**Soundness:** 2
**Presentation:** 2
**Contribution:** 2
**Rating:** 4
**Confidence:** 3

**Summary:**

The paper argues that under Non-IID data, dimensional collapse can occur at the classifier layer even if the backbone is robust. The authors propose FedRM, a multi-ratio random masking regularizer applied to both embeddings and the classifier during local training, which forces the head to exploit diverse feature dimensions rather than a few dominant directions. Experiments across several vision datasets and two backbones show consistent gains over FedAvg and several PEFT/zero-shot baselines, and the gains grow with dataset difficulty.

**Strengths:**

1. The paper articulates a specific and important problem within FLP. The concept of shifting "dimensional collapse" from the representation level to the classifier level is an interesting and plausible argument in the context of data heterogeneity.

2. The proposed FedRM method is conceptually clear and simple to implement.

3. The ablation study clearly demonstrates the benefit of using multi-ratio masks.

**Weaknesses:**

1. The paper's core premise is the existence of "classifier dimensional collapse". However, the evidence provided is indirect. The paper lacks a direct empirical analysis of this phenomenon, e.g., an analysis of the singular value spectrum or effective rank of the global classifier's weight matrix in the baseline (FedAvg FLP), which could visually demonstrate the "collapse" it claims to solve.

2. The idea of random masking as a regularizer has strong conceptual links to techniques like Dropout and DropConnect. The paper fails to compare or contrast FedRM with these classic regularization techniques.

3. The results in Appendix Table 5 show that FedRM's performance on the PathMNIST dataset (51.66%) is dramatically worse than the FedAvg baseline (65.49%). This "performance degradation"  directly contradicts the method's core claim of improving performance under heterogeneity. The authors note this in the text but provide no explanation.

**Questions:**

1. Can you provide direct evidence that "classifier dimensional collapse" is happening in the FedAvg (FLP) baseline and that FedRM is mitigating it?

2. Why does FedRM perform so much worse than the FedAvg baseline on PathMNIST (Table 5)? Does this imply that the regularization method can be detrimental for certain data distributions or tasks?

3. Regarding the ViT-Base results (Table 4), given that the Fed3R initialization is so effective, did you attempt to combine Fed3R initialization with your FedRM regularization? The two methods seem orthogonal and might yield the best performance.

---

### Official Review · Reviewer_TLQW · 2025-11-01

**Soundness:** 2
**Presentation:** 3
**Contribution:** 2
**Rating:** 2
**Confidence:** 3

**Summary:**

This paper developed an effective framework, namely FedRM, to mitigate classifier dimensional collapse under data heterogeneity. The core idea of FedRM is to employ multiple random masks with varying ratios applied to the embedding and the corresponding mask regularizations to jointly guide the optimization process. Extensive experimental results show the superiority of FedRM compared with existing baselines. However, there exist some concerns as follows: 1) The idea of this paper is too simple and why multiple random masks can address the issue of dimensional collapse is not clear; 2) A lot of federated pre-trained model frameworks were proposed recently, but the baselines used in the experiments are not state-of-art methods; 3) The theoretical analysis and experimental evaluation for why multiple random masks can address the issue of dimensional collapse are missing.

**Strengths:**

This paper developed an effective framework, namely FedRM, to mitigate classifier dimensional collapse under data heterogeneity. The core idea of FedRM is to employ multiple random masks with varying ratios applied to the embedding and the corresponding mask regularizations to jointly guide the optimization process. Extensive experimental results show the superiority of FedRM compared with existing baselines. The performance of FedRM becoming increasingly significant as the complexity of the dataset grows.

**Weaknesses:**

I have some concerns as follows:
1) The idea of this paper is too simple and why multiple random masks can address the issue of dimensional collapse is not clear;
2) A lot of federated pre-trained model frameworks were proposed recently, but the baselines used in the experiments are not state-of-art methods;
3) The theoretical analysis and experimental evaluation for why multiple random masks can address the issue of dimensional collapse are missing.

**Questions:**

I have some questions as follows:
1) The authors should add the theoretical analysis and experimental evaluation for why multiple random masks can address the issue of dimensional collapse are missing. The idea of this paper is too simple and why multiple random masks can address the issue of dimensional collapse is not clear.
2) A lot of federated pre-trained model frameworks were proposed recently. The authors should add some state-of-art methods in the experiments.
3) In some experimental results, the performance of FedRM looks poor, such as Tables 4 and 5.

---

### Note · Authors · 2025-11-13

I have read and agree with the venue's withdrawal policy on behalf of myself and my co-authors.